# LEARNING TO DEQUANTISE WITH TRUNCATED FLOWS

**Shawn Tan & Chin-Wei Huang**
Mila, University of Montreal
{jing.shan.shawn.tan,chin-wei.huang}@umontreal.ca

**Alessandro Sordoni**
Microsoft Research
Montreal
alsordon@microsoft.com

**Aaron Courville**
Mila, University of Montreal
Canada CIFAR AI Chair
courvila@iro.umontreal.ca

## ABSTRACT

Dequantisation is a general technique used for transforming data described by a discrete random variable $x$ into a continuous (latent) random variable $z$, for the purpose of it being modeled by likelihood-based density models. Dequantisation was first introduced in the context of ordinal data, such as image pixel values. However, when the data is categorical, the dequantisation scheme is not obvious. We learn such a dequantisation scheme $q(z|x)$, using variational inference with TRUncated FLows (TRUFL) — a novel flow-based model that allows the dequantiser to have a learnable truncated support. Unlike previous work, the TRUFL dequantiser is (i) capable of embedding the data losslessly in certain cases, since the truncation allows the conditional distributions $q(z|x)$ to have non-overlapping bounded supports, while being (ii) trainable with back-propagation. Addtionally, since the support of the marginal $q(z)$ is bounded and the support of prior $p(z)$ is not, we propose to renormalise the prior distribution over the support of $q(z)$. We derive a lower bound for training, and propose a rejection sampling scheme to account for the invalid samples. Experimentally, we benchmark TRUFL on constrained generation tasks, and find that it outperforms prior approaches. In addition, we find that rejection sampling results in higher validity for the constrained problems.

## 1 INTRODUCTION

Deep generative models aim to model a distribution of high-dimensional natural data. Many of these methods assume that the data is continuous, despite it being digitally stored in bits and therefore intrinsically discrete. This discrepancy has led to recent interest in *dequantising* discrete data types to avoid some of the degeneracies of fitting continuous models to discrete data (Theis et al., 2015). When data is ordinal (such as pixel intensities) a naive dequantisation scheme can be obtained by adding uniform noise to the discrete values (Theis et al., 2015). More recently, a generalisation of this approach where dequantisation is seen as inference in a latent variable model has also been proposed (Ho et al., 2019; Hoogeboom et al., 2020; Nielsen et al., 2020). However, these methods may not be directly applied in cases where the data is categorical (Hoogeboom et al., 2021), because the data is not naturally represented in a vector space.

Attempts at devising dequantisation schemes for categorical data by building upon the variational dequantisation scheme have been recently proposed in Hoogeboom et al. (2021) and Lippe & Gavves (2020). These approaches dequantise a categorical input into a latent continuous space. Ideally, a dequantisation scheme for categorical data should be: (i) easily learnable by standard optimization techniques and (ii) possibly lossless, in the sense that quantisation should recover the input category. Argmax Flow (Hoogeboom et al., 2021) offer lossless dequantisation but the support of the stochastic embedding is chosen arbitrarily and not optimised, and the dimensionality of the continuous (dequantised) variable is required to be at least logarithmic in the number of categories of the input data. Moreover, the method makes minimal assumptions about the topology of the categorical data, disregarding the possible relationships between categories, which can occur for example between word indices in natural language (Bengio et al., 2003) or the atomic representations

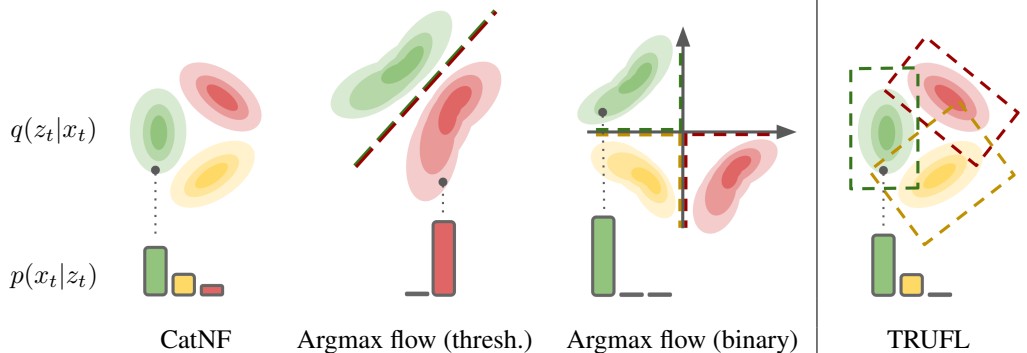

Figure 1: Decoder for CNFs (*left*) is the posterior of the encoder and the posteriors have unbounded support resulting in probability mass in all categories in $p(x_t|z_t)$. Argmax flow (*center*) have bounded supports that partitions the full latent space into regions where the posterior for each category is deterministic, but (1) the naive thresholding (thresh.) requires the dimensionality of the space to be equal to the number of categories, and (2) the binary encoding partitions data into separate octants, sometimes leaving octants unsupported. TRUFL (*right*) allows bounded support for each category, and for latent dimensions to be different from the category size.

of a molecule's constituents. On the other hand, Categorical Normalizing Flows (CatNF; Lippe & Gavves 2020) can learn a more compact representation of the input category but the dequantisation might be lossy given that the posteriors over the continuous variables have overlapping support.

Is there a trade-off between these two schemes? In this paper, we propose TRUFL, which builds upon the aforementioned variational dequantisation techniques. We achieve that by using truncated posterior distributions over the continuous variables with potentially bounded and disjoint support. In addition, we present a parametrisation of truncated distributions that can be optimised with standard stochastic reparametrisation techniques. Overall, our method inherits strengths of both CatNF and Argmax flow. Our experimental results highlight the effectiveness of our approach.

## 2 BACKGROUND: VARIATIONAL DEQUANTISATION

Dequantisation refers to the process of embedding discrete-valued data into a continuous space, which allows us to employ density-based models to capture the distribution of the continuous representation. Concretely, let $\boldsymbol{z} = \{z_1, \ldots, z_T\}$ denote this continuous representation, and $\boldsymbol{x} = \{x_1, \ldots, x_T\}$ describe the observed data, where each $x_t$ represent, *e.g.* a node in a graph or a token in a sentence. Each $x_t$ is assumed to be categorical, i.e. $x_t \in \{0, \cdots, K-1\}$ for some integer $K > 1$. $\boldsymbol{z}$ can be interpreted as a latent variable, which follows a prior distribution $p(\boldsymbol{z})$. We refer to $q(z_t|x_t)$ as the *dequantiser* and $p(x_t|z_t)$ as the *quantiser*. Training can be achieved by maximizing a variational lower bound on the marginal likelihood of the data, *i.e.*:

$$\log p(\boldsymbol{x}) \geq \mathbb{E}_{q(\boldsymbol{z}|\boldsymbol{x})}\left[\log \frac{p(\boldsymbol{x}|\boldsymbol{z})\,p(\boldsymbol{z})}{q(\boldsymbol{z}|\boldsymbol{x})}\right] =: \mathcal{L}(\boldsymbol{x}) \tag{1}$$

We are interested in the case where the representation $z_t$ can be inferred from $x_t$ alone, so we choose the factorisation $p(\boldsymbol{x}|\boldsymbol{z}) = \prod_t p(x_t|z_t)$ and $q(\boldsymbol{z}|\boldsymbol{x}) = \prod_t q(z_t|x_t)$, following Lippe & Gavves (2020). In this case, the "optimal" quantiser $p(x_t|z_t)$ can be conveniently computed as:

$$\operatorname*{arg\,max}_{p(x_t|z_t)} \mathbb{E}_{q(\boldsymbol{x})}[\mathcal{L}(\boldsymbol{x})] = \frac{q(z_t|x_t)\,\tilde{p}(x_t)}{\sum_{x_t'=0}^{K-1} q(z_t|x_t')\,\tilde{p}(x_t')} = q(x_t|z_t) =: p(x_t|z_t) \tag{2}$$

where $q(\boldsymbol{x})$ denotes the (empirical) data distribution, and $\tilde{p}(x_t)$ denotes the estimate of the marginal distribution of each category (which can be obtained by counting and, in the case of textual data, this corresponds to the unigram distribution over words). This equation shows that the optimal quantiser can be obtained implicitly by applying Bayes' rule with the parametric dequantiser $q(z_t|x_t)$. The factorisation we chose for $p(\boldsymbol{x}|\boldsymbol{z})$ and $q(\boldsymbol{z}|\boldsymbol{x})$ is crucial for the $\arg\max$ above to be represented in this

simple form. Without this assumption, the solution will involve a combinatorial sum or an integral, which results in the choice of a parametric quantiser in Ziegler & Rush (2019) for computational tractability. Plugging the optimal decoder into Eq. 1 yields:

$$\mathcal{L}(\boldsymbol{x}) = \mathbb{E}_{q(\boldsymbol{z}|\boldsymbol{x})} \left[ \sum_t \log \tilde{p}(x_t) + \log \frac{p(\boldsymbol{z})}{\sum_{x'_t=0}^{K-1} q(z_t|x'_t)\tilde{p}(x'_t)} \right] \tag{3}$$

We note that the first term is a constant. Therefore, the expression above implies that accurately modelling the dependencies in $\boldsymbol{x}$ boils down to learning an expressive prior $p(\boldsymbol{z})$ and regularising the dequantiser $q(z_t|x_t)$. $q(x_t|z_t)$ is deterministic when $q(z_t|x_t)$ does not overlap with other $q(z_t|x'_t)$, in which case $q(z_t|x_t)$ is encouraged to be expanded to maximize the entropy. If there is certain amount of overlapping, the denominator in the second term will push down the density of other $q(z_t|x'_t)$, therefore resulting in a spikier aggregate posterior distribution (see more discussion on this in Section 5.3). With this general framework that also accounts for lossy quantisation, we briefly present some of the previously proposed strategies for dequantisation.

**Ordinal dequantisation**   In the case where the data is ordinal, such as the case of image pixel values (*e.g.* for an 8-bit representation, $K = 256$), a dequantisation scheme can be obtained by setting $q(z_t|x_t) = \text{Uniform}(x_t, x_t + 1)$. The resulting quantisation process is simply $\lfloor z_t \rfloor$, and is deterministic. More generally, $q(z_t|x_t)$ can be any distribution on $[x_t, x_t+1]$. See Nielsen & Winther (2020); Hoogeboom et al. (2019) for extensions of the uniform dequantisation scheme.

**Argmax Flow**   For categorical data, uniform dequantisation is not applicable, as there is no intrinsic ordering between the categories. Argmax Flow (Hoogeboom et al., 2021) dequantise categorical data by letting $z_t \in \mathbb{R}^K$ be distributed by $q(z_t|x_t)$ with support $\{z_t : \arg\max_k(z_t)_k = x_t\}$. When the support over the latent space is disjoint, $p(x_t|z_t) = q(x_t|z_t) = \mathbf{1}[x_t = \arg\max_k(z_t)_k]$[1]; we depict this in Figure 1, Argmax flow (thresh.). Argmax Flow makes minimal assumptions on the topology of the data: the support of the dequantiser partitions the continuous space evenly and the representations are equally far from each other. As an example, synonyms in text may still have very distinct dequantised representations despite having similar functions and meaning in a language modelling setting. In the naive formulation, Argmax Flow requires the dimensionality to the latent space to be the same as the number of the input categories $K$. To accomodate for larger categorical spaces, the authors suggest a binary factorisation, reducing the required latent space dimension to $\lceil \log_2 K \rceil$; See Figure 1, Argmax flow (binary).

**Categorical Normalising Flows (CatNF)**   In the previous cases, the quantisation is deterministic, and there is no loss of information. This is because in a cleanly partitioned latent space like in the ordinal setting or argmax flow, the dequantising distributions $q(z_t|x_t = k)$ for all $0 \leq k \leq K - 1$ have non-overlapping support. CatNF learns a dequantiser that can "softly" partition the space. Lippe & Gavves (2020) propose using a conditional logistic distribution as $q(z_t|x_t)$. In this case, the optimal quantiser $q(x_t|z_t)$ is nearly-deterministic if the locations of the dequantisers are far away from each other and they have sufficiently small scale. For this reason, and unlike the first two approaches, CatNF is not capable of losslessly dequantising the data (we provide a formal discussion on this in Appendix A.2, which is based on an data-processing inequality argument, using the dequantiser as a transitional kernel). It can approximate the lossless limit by pushing the bulk of the mass of $q(z_t|x_t)$ away from each other, but that could potentially lead to a highly complex and multi-modal empirical distribution over the representation space for $p(\boldsymbol{z})$ to approximate.

Next, we consider the case where $q(z_t|x_t)$ is a truncated distribution, and as such has the ability to encode the data losslessly while learning a meaningful latent topology of the different categories.

## 3   TRUNCATED DEQUANTISER AND PRIOR

The general approach to optimising the variational lower bound proposed in Kingma & Welling (2014) involves sampling from the proposal distribution $q(z_t|x_t)$ to estimate the expectation (Eq. 1). In our case, we want to parameterise this using a TRUncated FLow, which we will refer to as TRUFL.

---

[1]We use $\mathbf{1}[\cdot]$ to denote an indicator function.

For simplicity, we will drop the dependency on $t$ and $x_t$, but all of the variational distribution is conditioned on the categorical value $x_t$.

If we want to bound a scalar distribution between $(a, b)$, and we have a density function $f$ where its cumulative distribution function $F$ (CDF) and the inverse of its CDF $F^{-1}$ are tractable, we can easily sample from $f$ by sampling $u$ from $\text{Uniform}(F(a), F(b))$, and then evaluating $F^{-1}(u)$. Note that this method is differentiable, and we use it to sample from our dequantiser.

However, multi-variate distributions may not simply be truncated at the tails, but rather have a support which is a strict subset of its base distribution. One general approach to sampling from such a distribution is via rejection sampling (Murphy, 2012). This approach has been used in prior work for sampling from bounded-support distributions (Polykovskiy & Vetrov, 2020; Xu & Durrett, 2018; Davidson et al., 2018). Computing gradients for this method is possible via implicit gradients (Figurnov et al., 2018), but we do not need gradients in our case, as we use rejection sampling for generating samples from the generative model (See Section 3.2).

## 3.1 TRUNCATED LOGISTIC DISTRIBUTION

We choose the truncated logistic distribution to parameterise the dequantiser, because the density, the CDF and the inverse CDF of the logistic distribution can all be easily computed:

$$f(z) = \sigma(z) \cdot \sigma(1-z), \qquad F(z) = \sigma(z), \qquad F^{-1}(u) = \log \frac{u}{1-u}, \tag{4}$$

for $z \in \mathbb{R}$ and $u \in (0, 1)$, where $\sigma$ is the logistic sigmoid function. Additionally, we can parameterise the width of $\text{Uniform}(F(a), F(b))$ directly by $s$, where $0 < s \leq 1$, and the mean of this distribution as $m$, such that $m + \frac{s}{2} < 1$ and $m - \frac{s}{2} > 0$. Given a set of statistical parameters $\hat{s} \in \mathbb{R}$ and $\hat{m} \in \mathbb{R}$ for each category, we can impose these constraints by the following reparameterisation:

$$m = \sigma(\hat{m}), \qquad s = 2 \cdot \min(m, 1-m) \cdot \sigma(\hat{s}). \tag{5}$$

Then, to sample from $\text{Uniform}(F(a), F(b))$,

$$u = m + \left(u_0 - \frac{1}{2}\right) \cdot s, \qquad u_0 \sim \text{Uniform}(0, 1). \tag{6}$$

This allows for a simple implementation for a truncated logistic distribution that is differentiable (see Appendix A.1 for the derivation of the gradient). We can write the pdf as

$$\tilde{f}(z; m, s) = \begin{cases} \frac{1}{s}\sigma(z) \cdot \sigma(1-z), & \text{if } 0 < \frac{F(z)-m}{s} + \frac{1}{2} \leq 1 \\ 0, & \text{otherwise.} \end{cases} \tag{7}$$

Like in Lippe & Gavves (2020), the resulting approximate posterior $q(z_t|x_t)$ requires category specific parameters for the truncation ($m(x_t)$ and $s(x_t)$) and any flow applied on top of the truncated logistic (we denote this flow as $g$). During training, conditioned on a given category $x_t$, we can sample $z_t$ with the method described above, and compute the probability $q(z_t|x_t)$. Since decoding requires computing $g^{-1}(z_t, \hat{x}_t)$ for all $K$ categories, it limits the choice of flows applicable. In CatNF implementations, a linear flow is used (Kingma & Dhariwal, 2018), while empirically the authors find that a category conditional scale and shift suffices. Algorithm 1 details the steps taken for computing $z_t, \log q(z_t|x_t)$, and $\log p(x_t|z_t)$.

With this parameterisation of the decoder and the truncated approximate posterior, some $q(z_t|\hat{x}_t)$ where $x_t \neq \hat{x}_t$ may not have support over the sample $z_t$. In the extreme case, when $q(z_t|x_t)$ for all possible assignments of $x_t$ had mutually exclusive support, then decoding will always be deterministic, similar to Argmax flow. This is when TRUFL has the flexibility to embed discrete data losslessly in a continuous space. Empirically we find that $q(x_t|z_t)$ is sparse, but not always deterministic.

**Recovering Argmax Flow and CatNF** Specific choices of $m$ and $s$ in Eq. 6 can recover CatNF and Argmax Flow support for each category. By setting $m = 0.5$ and $s = 1$, we perform no truncation on the posterior distribution, where we recover CatNF. For Argmax Flow, we can achieve orthant support by setting $m$ to 0.25 or 0.75 for each dimension according to the orthant it was assigned based on the binary encoding, and $s = 0.25$. This will ensure that the support covers only half of the real line in that dimension, thus matching Argmax Flow support for that category.

---

**Algorithm 1** Truncated Categorical Encoding for a timestep $t$

---

**Input:** Categorical data $x_t$, Flow $g(\cdot, \cdot)$
**Output:** $z_t, \log q(z_t|x_t), \log p(x_t|z_t)$
$u_0 \sim \text{Uniform}(0, 1)$                                    $\triangleright$ Begin encoding
$u \leftarrow m(x_t) + \left(u_0 - \frac{1}{2}\right) \cdot s(x_t)$
$z'_t \leftarrow F^{-1}(u)$
$z_t \leftarrow g(z'_t, x_t)$                                        $\triangleright$ End Encoding
**for** $\hat{x}_t = 0$ to $K - 1$ **do**                  $\triangleright$ Compute probability of $z_t$ given all possible $\hat{x}_t$
    $\hat{z}'_t \leftarrow g^{-1}(z_t, \hat{x}_t)$
    $\log q(z_t|\hat{x}_t) \leftarrow \log \tilde{f}(\hat{z}'_t; m(\hat{x}_t), s(\hat{x}_t)) + \log \left|\frac{\mathrm{d}z'_t}{\mathrm{d}\hat{z}_t}\right|$
**end for**
$\log p(x_t|z_t) \leftarrow \log q(z_t|x_t)\,\tilde{p}(x_t) - \log \sum_{\hat{x}_t} q(z_t|\hat{x}_t)\,\tilde{p}(\hat{x}_t)$      $\triangleright$ log computation of the $q$ posterior

---

**Algorithm 2** Rejection sampling

---

**Output:** $x$
$z \sim p(z)$
**while** $\forall \hat{x} : \tilde{f}(z; m(\hat{x}), s(\hat{x})) = 0$ **do**
    $z \sim p(z)$
**end while**
$x \sim p(x|z)$

---

$p(z)$                  $p_{\text{trunc}}(z)$

Figure 2: *Left:* Pseudo-code for rejection sampling. *Centre:* The untruncated prior $p(z)$. *Right:* The prior truncated according to the unioned support of $q$. We denote this as $p_{\text{trunc}}(z)$.

## 3.2 REJECTION SAMPLING

When sampling from the model, the standard process is ancestral sampling: $z \sim p(z)$, and then $x \sim p(x|z)$. However, in our model, the sample $z_t$ may not have support under any of the $q(z_t|x_t)$, resulting in an undefined optimal $p(x_t|z_t)$. As mentioned, we reject the samples that have no support for any of the mixture components. This is equivalent to redefining a new prior $p_{\text{trunc}}(z)$ proportional to $p(z)$ such that its support matches the unioned support of $q(z|x)$ and renormalising. The existing ELBO is still a valid lower-bound of such a distribution:

$$\mathcal{L} = \mathbb{E}_{q(z|x)}\left[\log \frac{p(x|z)\,p(z)}{q(z|x)}\right] \leq \mathbb{E}_{q(z|x)}\left[\log \frac{p(x|z)\,\frac{p(z)}{Z}}{q(z|x)}\right] \leq \int p(x|z)\,p_{\text{trunc}}(z)\,\mathrm{d}z, \quad (8)$$

$$\text{where} \qquad Z := \int p(z) \cdot \mathbf{1}\left[\exists \hat{x} : \tilde{f}(z; m(\hat{x}), s(\hat{x})) > 0\right]\,\mathrm{d}z \leq 1. \quad (9)$$

Eq. 9 in principle can be estimated by counting the frequency of accepted samples. For one of our experiments, the overall rejection rate at the end of training is about 0.14, suggesting that rejection sampling from a trained prior is not inefficient (see Section 5.1). Using the decoder as an accepting heuristic, we find empirically that this results in more valid samples in constrained tasks that require sampling from the model.

## 4 RELATED WORK

**Flows on Discrete Data** Discrete flows (Tran et al., 2019; Hoogeboom et al., 2019) deal with flows in the discrete space, but because of this they resort to the straight-through method (Bengio et al., 2013) for estimating gradient. Using conditional permutations, Lindt & Hoogeboom (2021) develop a way to use discrete flows that do not have the same gradient bias that Discrete flows introduce due to its gradient estimator. Our approach maps the discrete data into a continuous space, allowing the prior to take on most of the modelling complexity.

Table 1: Results on the graph coloring problem (Lippe & Gavves, 2020). SMALL are graphs of size $10 \leq |V| \leq 20$, and LARGE $25 \leq |V| \leq 50$. All results are attained using the CatNF codebase, and averaged across 3 random seeds. Results in the rounded box are using a different set of hyperparameters than the ones used in CatNF.

| | SMALL | | LARGE | |
|---|---|---|---|---|
| **Method** | **Validity** | **bpd** | **Validity** | **bpd** |
| RNN+Largest first | 93.41% ±0.42% | 0.68 ±0.01 | 71.32% ±0.77% | 0.43 ±0.01 |
| CatNF | 94.56% ±0.55% | 0.67 ±0.00 | 66.80% ±1.14% | 0.45 ±0.01 |
| | | | 68.06% ±0.04% | 0.45 ±0.00 |
| Argmax (thresh.) | 94.81% ±0.37% | 0.66±0.00 | 63.65% ±0.24% | 0.46 ±0.00 |
| No rejection sampling | 95.40% ±0.35% | 0.65 ±0.01 | 68.10% ±0.00% | 0.45 ±0.00 |
| Rejection sampling | **95.90%** ±0.29% | | **74.20%** ±0.01% | |

**Learning the Prior**   Since most of the modelling happens in the prior after dequantisation, the prior has to have the expressibility to model the resulting continuous data. Learnable priors are not new in the VAE literature (Tomczak & Welling, 2018; Huang et al., 2017). One common way for learning sequential data is to use an autoregressive prior (Ziegler & Rush, 2019). Liu et al. (2019) introduces a graph-structured flows which can be used for modelling such data, and there have been flow-based models developed for molecule generation (Madhawa et al., 2019; Shi et al., 2020). Ziegler & Rush (2019) and Huang et al. (2018) also introduce normalising flows that can learn multi-modal target distributions, which Ziegler & Rush (2019) show is important in modelling text data.

**Optimal decoder**   In the case of categorical data with a sufficiently small $K$, computing the posterior of the encoder is tractable during training. The optimal decoder in Eq. 3 was used in practice in Lippe & Gavves (2020). Argmax flow (Hoogeboom et al., 2021) are a special case in which the encoding of the categorical variable results in a latent variable that has a deterministic decoding, which Nielsen et al. (2020) generalises and calls *surjectivity* which can be parameterised in either inference or generation. The optimal decoder also has the added effect of alleviating the posterior collapse problem commonly faced when modelling discrete data such as text (Yang et al., 2017; Bowman et al., 2015; Ziegler & Rush, 2019; Chen et al., 2016).

**Bounded-support distributions in latent variable models**   While normalising flows (Papamakarios et al., 2019) allow for learning a more flexible posterior distribution, Jaini et al. (2020) and Verine et al. (2021) discuss the current limitations of flows due to their bi-lipschitz property which result in nearly no transformation at the tails of distributions. Truncated distributions are an extreme case of a light-tailed distribution, achieved by performing a shift and scaling of the base uniform distribution. There have also been efforts on spherical latent variables, modelled by a von Mises-Fisher (vMF) distribution (Xu & Durrett, 2018; Davidson et al., 2018). Polykovskiy & Vetrov (2020) uses bounded-support kernels as the basis of their posterior distributions, and modify the decoder so that the results are deterministic, but introduce gradient estimations in order to train the model. In the latter two cases, rejection sampling was used in order to attain samples for inference.

## 5 EXPERIMENTS

In this section, we present experiments on TRUFL. See Appendix C for implementation details.

### 5.1 GRAPH COLORING

We first consider the graph colouring problem (Bondy et al., 1976) introduced in Lippe & Gavves (2020). The task is to colour the nodes in a graph with 3 different colours such that any two nodes connected by an edge do not have the same colour, which provides a meaningful benchmark for evaluating how well a model learns unknown constraints from the data. We evaluate the **Validity** of the generated samples conditioned a given graph.

Table 2: Performance on molecule generation trained on Zinc250k (Irwin et al., 2012), calculated on 10k samples and averaged over 4 random seeds.

| Method | Validity | Uniqueness | Novelty |
|---|---|---|---|
| *Constrained generation* | | | |
| JT-VAE (Jin et al., 2018) | 100% | 100% | 100% |
| GraphAF (Shi et al., 2020) | 68% | 99.10% | 100% |
| R-VAE (Ma et al., 2018) | 34.90% | 100% | — |
| GraphNVP (Madhawa et al., 2019) | 42.60% | 94.80% | 100% |
| Argmax flow (thresh.) | 78.36% $\pm$4.93% | 100% | 99.99% |
| CNF (Lippe & Gavves, 2020) | 83.41% $\pm$2.34% | 99.99% | 100% |
| $\hookrightarrow$ Sub-graphs | 96.35% $\pm$2.58% | 99.98% | 99.98% |
| No rejection sampling | 84.46%$\pm$4.24% | 100% | 100% |
| $\hookrightarrow$ Sub-graphs | 97.51%$\pm$2.37% | 99.98% | 99.99% |
| $\hookrightarrow$ Rejection sampling | 85.01%$\pm$1.05% | 100% | 100% |
| $\hookrightarrow$ Sub-graphs | **97.87%**$\pm$0.06% | 100% | 100% |

For baselines, we present results from CatNF (Lippe & Gavves, 2020), and implemented the thresholding version of Argmax flow. We implement the truncated flows using the CatNF codebase, and compare results with and without rejection sampling for TRUFL. Additionally, we find a better set of hyperparameters work better for the LARGE setting, and report the results base on those hyperparameters for the baseline as well. All graph-structured flows used in this task are the same as the ones detailed in CatNF, we simply replace the categorical embedding module. For results without rejection sampling, we also compute the probability over the categories using Eq. 9, but disregard the truncation of the densities. We find we perform marginally better than the baselines in the small graph regime. However, with rejection sampling and better hyperparameters, we significantly outperform the flow-based benchmarks, and perform better than the RNN baseline used in Lippe & Gavves (2020).

**Rejection Rate of TRUFL** Since the prior will be optimised to fit the aggregate posterior during training, we expect the rejection rate of sampling from the truncated prior to drop as training progresses. We measure the rejection rate when sampling from the model and report its average rate of rejection as training progresses. While the model is conditioned on graphs of different sizes during sampling, we look at the average rate of rejection go obtain a measure of the efficiency of our rejection sampling scheme. Figure 3 suggests that the prior learns to fit the support of the aggregate posterior better during training, as the rejection rate reduces from about 0.45 to about 0.14 at the end of training. This

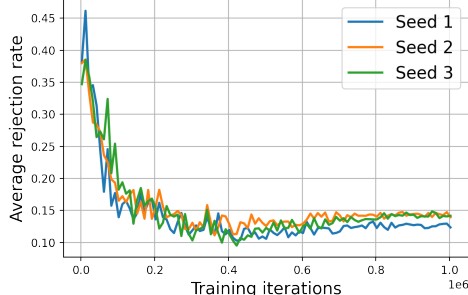

Figure 3: Rejection rate vs training iterations. The rejection rate when sampling for evaluation for the graph colouring task reduces during iteration, indicating (1) the rejection rate is not intractably high, and (2) the prior tries to match the support of TRUFL over time.

corresponds to roughly losing $\log Z \approx 0.15$ nats (or 0.22 bits) per molecule. Note that we did not account for this gap while computing the bpd in Table 1 since we would have to estimate $\log Z$ conditioned on each graph size, and this would give us slightly lower value in theory.

## 5.2 MOLECULE GENERATION

Molecule generation is another task commonly used as a generative modelling benchmark (Jin et al., 2018; Shi et al., 2020; Ma et al., 2018; Madhawa et al., 2019). The atoms constituting the molecules are naturally represented by tokens that are usually interpreted categorically, but the intrinsic structure of the atoms, such as the electron configurations, is often neglected. For this reason, we believe TRUFL and CatNF will learn a more useful representation than Argmax Flow. Moreover, given the

Table 3: Results on character-level and word-level language modelling, an average across 3 different random seeds. Results reported with † indicates results attained with the Argmax flow codebase. ∗ indicates results attained with the CatNF codebase. All other results are from the CatNF paper.

| | CHARACTER | | WORD |
|---|---|---|---|
| **Model** | **PTB (char.)** | **Text8** | **Wikitext103** |
| LSTM | 1.28 ±0.01 | 1.44 ±0.01 | 4.81 ±0.05 |
| Latent NF (Ziegler & Rush, 2019) | 1.30 ±0.01 | 1.61 ±0.02 | 6.39 ±0.19 |
| Categorical NF (Lippe & Gavves, 2020) | 1.27 ±0.01 | 1.45 ±0.01 | 5.43 ±0.09 |
| Argmax Flow (Hoogeboom et al., 2021) | *$\mathbf{1.26}$ ±0.01 | †$\mathbf{1.39}$ ±0.00 | *5.42 ±0.01 |
| Ours | *$\mathbf{1.26}$ ±0.02 | †1.40 ±0.01 | *$\mathbf{5.35}$ ±0.01 |

Table 4: Results on word-level language modelling, each sentence is modelled as a separate data point. CatNF, Argmax Flow, and TRUFL results are averaged across 4 random seeds. Setting and baseline LSTM provided by Kim et al. (2019).

| Model | Dim. | PPL | | NLL | | Recon. | KL |
|---|---|---|---|---|---|---|---|
| LSTM | — | 86.2 | | 4.46 | | — | — |
| CatNF | 12 | 139.7 | ± 3.0 | 4.93 | ± 0.02 | 1.01 | 4.18 |
| Argmax Flow (binary) | 14 | 242.7 | ± 2.7 | 5.49 | ± 0.01 | 0.00 | 5.75 |
| Ours | 12 | 143.6 | ± 4.9 | 4.97 | ± 0.03 | 1.47 | 3.71 |

results in graph colouring, we believe that rejection sampling can aid in creating more valid samples than prior unconstrained generation methods. Molecule generation also requires the generation of edge categories in addition to the node categories.

Our benchmarks were implemented with the Lippe & Gavves (2020) codebase, and the Argmax flow benchmark uses the same thresholding scheme. Rejection sampling in this case requires rejecting invalid edges and vertices both, and so resulted in longer sampling times. The results are in Table 2 Again, we find that TRUFL consistently improves upon CatNF in terms of the validity of the generated samples, which can be further increased by truncating the prior via rejection sampling, and taking the largest sub-graph in cases where multiple disconnected graphs are generated (as in done in Lippe & Gavves (2020)).

## 5.3 LANGUAGE MODELLING

One can view these variational dequantisation methods when applied to language modelling as a method of using *stochastic embeddings*: each token is represented by a *distribution* over the embedding space, and language models then operate over samples from these distributions. Probabilistic embeddings have been suggested in prior work (Li et al., 2018; Dasgupta et al., 2020; Chen et al., 2021), but here, distribution parameters are trained with a language modelling objective. We perform experiments on character-level and word-level language modelling. For word-level language modelling, many tasks require a large number of categories (∼10k). The naive thresholding method will result in a large latent space, no different from using a one-hot representation, so we use the previously mentioned binary encoding of each category in our implementation of the Argmax flow.

For the character level experiments on the Penn Treebank (PTB; Marcus et al. 1993), we use the setup in CatNF, replacing the categorical encoding with TRUFL. For text8 (Mikolov et al., 2014), we use the Argmax flow codebase, modifying the Argmax flow module and replacing it with TRUFL. For word level experiments on Wikitext103 (Merity et al., 2016), we use the setting provided by CatNF as well, which evaluates chunks of 256 words, and initialises the word embeddings using GloVe (Pennington et al., 2014). While this is not the standard setting for language modelling, we include these results for comparison with Argmax Flow and CatNF. We report the negative log-likelihood in Table 3. For word level experiments on PTB, we use the setting specified by Kim et al. (2019), which also provides an LSTM baseline. Unlike standard PTB results, this benchmark treats each sentence

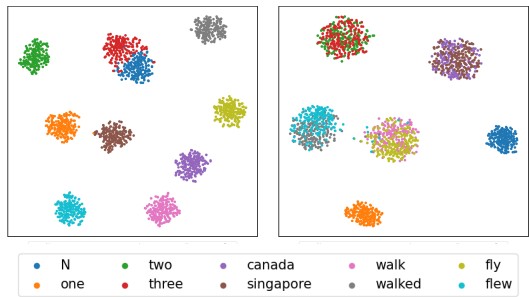

Figure 4: Scatter plot of t-SNE embeddings of samples from Armgax Flows *(left)* and TRUFL *(right)*. Since t-SNE reveals the relative proximities of the embeddings, we should note that the visualisation here reveals what clusters of embeddings are close to others *relative* to the other words in the plot.

as i.i.d., instead of treating the entire dataset as a continuous string of text. We provide further details of the architecture in Appendix C.3.

The reconstruction loss for TRUFL is higher than that of CatNF, suggesting that CatNF may have optimised the variance of each dequantising distribution to be fairly small in order to approach deterministic decoding. To verify this, we compute the Within-Group Standard Deviation (WGSD) across all $q(z_t|x_t)$. Sampling 200 samples from each category, we first normalise the mean and standard deviation across all $10,000 \times 200$ categories and samples. We then compute the standard deviation for each category across the 200 samples, and average across all 12 dimensions. CatNF has an WGSD of **0.52**, while TRUFL has an WGSD of **0.66**. A smaller WGSD would indicate peakier disributions in the latent space, given that each quantising distribution has a lower dispersion on average. Since the prior is modelling the correlations between the categories in the sequence, a peakier distribution will require more capacity to model. In this scenario, we find that both models are comparable, with CatNF performing slightly better. However, recall that even with importance sampling, we are not accounting for the unsupported regions of the prior. This means this is an estimate of the upper bound in Eq. 8, and that estimating $Z$ will result in a lower perplexity score. In general, there has been a gap between autoregressive models (*e.g.* RNNs, Transformers) that model the discrete distribution directly, and latent variable models, and we believe this gap will close as dequantising techniques improve.

We perform a qualitative analysis on the learned stochastic embeddings, comparing the t-SNE plots of samples of 10 chosen words. We took 200 samples from $q(z_t|x_t)$ for each word, and performed t-SNE over all $200 \times 10$ vectors. As expected, Argmax flow partitions the latent space into separate orthants, resulting in word embeddings that are arbitrarily chosen. N and `three` are an exception, but this is perhaps due to a chance occurrence of their orthants being dissimilar in only one dimension. On the other hand, TRUFL learning the topology of these embeddings results in clusters that retain some properties of the words they represent. Interestingly, while `flew:fly` and `walked:walk` are different tenses of the same word, the learned distribution for each word appears to group them by tense instead. Since this would inform predictions of the tense of future words, this is perhaps a functional representation of these words.

## 6 CONCLUSION

In this paper, we propose TRUFL, a flow based dequantiser with truncated support for stochastically embedding categorical data in a continuous space. This allows us to employ density-based models like normalising flows to fit the continuous representation of the data. To deal with the unsupported regions in the decoder, we further propose truncating the prior distribution to account for the unioned support of the dequantiser, which proves to be more effective at generating samples with constraints, such as for graph coloring and molecule generation. In language modelling, we find that learned stochastic embeddings has better performance than a strict orthant partitioning of the space, and qualitative analysis reveals that the proximity between similar words under such a partitioning is poor. We believe that further work on variational dequantisation will close the gap between auto-regressive models that directly model the categorical space and latent variable models.

ACKNOWLEDGEMENTS

We would like to thank Yikang Shen and Christos Tsirigotis for their input and suggestions during the course of this work. Phillip Lippe was also gracious in helping us understand the specifics of the CatNF codebase and running the experiments.

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

## A  APPENDIX

### A.1  GRADIENT ESTIMATES OF $a$ AND $b$

$$p(z; a, b) = \begin{cases} \frac{1}{b-a}, & \text{if } a < z < b, \\ 0, & \text{otherwise} \end{cases} \tag{10}$$

Let

$$z = a + (b - a) \cdot u \tag{11}$$
$$u \sim \text{Uniform}(0, 1) \tag{12}$$
$$u = \frac{z - a}{b - a} \tag{13}$$

$$\nabla_a \mathbb{E}_z[f(z)] = \mathbb{E}_{u \sim \text{Uniform}(0,1)} [\nabla_a f(a + (b - a) \cdot u)] \tag{14}$$
$$= \mathbb{E}_{u \sim \text{Uniform}(0,1)} [(1 - u)(\nabla_z f(z))] \tag{15}$$
$$\nabla_b \mathbb{E}_z[f(z)] = \mathbb{E}_{u \sim \text{Uniform}(0,1)} [u(\nabla_z f(z))] \tag{16}$$

### A.2  SOFT DEQUANTIZATION

Let $x$ denote the data (*e.g.* a sentence) and $L$ denote the dimensionality of the data; *i.e.* $x \in \mathcal{C}^L$ and $L \in \mathbb{Z}_+$. $x$ can be dequantized by sampling $z_j \sim q(z_j|x_j)$ for $j \in 1, ..., L$. When $\{q(z_j|x_j = c) : c \in \mathcal{C}\}$ has the same support, we call it *soft* dequantization. We write $p(x_j|z_j)$ and $p(z)$ to denote the quantizer and the prior over the dequantized data. We consider the following ELBO

$$\mathcal{L}(x, L) = \mathbb{E}_{\prod q(z_j|x_j)} \left[ \sum_{j=1}^{L} \log \frac{p(x_j|z_j)}{q(z_j|x_j)} + \log p(z) \right] \tag{17}$$

**Theorem 1** (Maximizer of ELBO, dequantization). *Assume $q(x, L) > 0$ for all $x \in \mathcal{C}^L$ and $L \in \mathbb{Z}_+$, and that $x_1, ..., x_L$ are not conditionally independent given $L$. Then*

$$\mathbb{E}_{q(x,L)}[\log q(x|L)] > \mathbb{E}_{q(x,L)}[\mathcal{L}] \tag{18}$$

*as long as the densities $q(z_j|x_j = c)$ for different categories $c \in \mathcal{C}$ have the same support. That is, soft dequantization with shared support is suboptimal.*

*Proof.* The optimal quantizer is

$$q(x_j|z_j) \propto q(x_j)q(z_j|x_j) \quad \text{where} \quad q(x_j) \propto \sum_{L=1}^{\infty} \sum_{j'=1}^{L} q(x'_j = x_j, L) \tag{19}$$

Plugging it into the ELBO gives

$$\mathcal{L}(x, L) = \mathbb{E}_{\prod q(z_j|x_j)} \left[ \sum_{j=1}^{L} \log \frac{q(x_j|z_j)}{q(z_j|x_j)} + \log p(z) \right] = \mathbb{E}_{\prod q(z_j|x_j)} \left[ \sum_{j=1}^{L} \log q(x_j) + \log \frac{p(z)}{\prod_j q(z_j)} \right] \tag{20}$$

where $q(z_j) = \sum_{x'_j} q(x'_j)q(z_j|x'_j)$.

On the other hand, we can rewrite the negentropy of the data as

$$\mathbb{E}_{q(x,L)}[\log q(x|L)] = \mathbb{E}_{q(x,L)} \left[ \sum_{j=1}^{L} \log q(x_j) + \log \frac{q(x|L)}{\prod_{j=1}^{L} q(x_j)} \right] \tag{21}$$

Now to compare the second term with that of the ELBO we let $q(z|L) = \sum q(x|L) \prod_{j=1}^{L} q(z_j|x_j)$. For simplicity, we denote by $T(z|x) = \prod_{j=1}^{L} q(z_j|x_j)$ the transition kernel applied to either $q(x|L)$ or $\prod_{j=1}^{L} q(x_j)$. Then

$$\mathbb{E}_{q(x,L)}\left[\log \frac{q(x|L)}{\prod_{j=1}^{L} q(x_j)}\right] = \mathbb{E}_{q(x,L)T(z|x)}\left[\log \frac{q(x|L)T(z|x)}{\left(\prod_{j=1}^{L} q(x_j)\right)T(z|x)}\right] \tag{22}$$

$$= \mathbb{E}_{q(z,L)q(x|z,L)}\left[-\log \frac{\left(\prod_{j=1}^{L} q(x_j)\right)T(z|x)}{q(x|L)T(z|x)}\right] \tag{23}$$

$$\overset{*}{>} \mathbb{E}_{q(z,L)}\left[-\log\left(\mathbb{E}_{q(x|z,L)}\left[\frac{\left(\prod_j q(x_j)\right)T(z|x)}{q(x|L)T(z|x)}\right]\right)\right] \tag{24}$$

$$= \mathbb{E}_{q(z,L)}\left[-\log \frac{\prod_j q(z_j)}{q(z|L)}\right] = \mathbb{E}_{q(z,L)}\left[\log \frac{q(z|L)}{\prod_j q(z_j)}\right] \tag{25}$$

which is greater or equal to the second term of the ELBO by Gibb's inequality. The inequality marked by $*$ is due to Jensen and the convexity of $-\log$, and is strict since the dependency of $x_{1,...,L}$ implies there exist some $x^1$ and $x^2$ s.t. $\prod_j q(x_j^i) \neq q(x^i|L)$ for $i \in \{1, 2\}$. Now, since $T(z|x) > 0$ over the shared support by assumption, for almost all $z$ and $L$, $q(x^i|z, L) > 0$ for $i \in \{1, 2\}$. That is, with non-zero probability the argument to the convex function $-\log$ has a different value from its mean. This implies the inequality is strict and concludes the proof.

$\square$

## B  FURTHER RELATED WORK

**Consequences of a binary encoding**  The binary encoding scheme of Argmax flow bears some resemblance to prior work on hierachical softmax schemes to reduce the computational footprint of the large softmax layer (Goodman, 2001; Morin & Bengio, 2005; Mnih & Hinton, 2008; Shen et al., 2017). Specifically, Mnih & Hinton (2008) proposed an analogous scheme for scaling up the softmax layer by representing words as leaves in a tree, and binary encodings as a traversal of such a tree. However, the partitioning of the $\lceil \log_2 K \rceil$-dimensional space into $K$ equal categories will only be exact if $K$ is a power of 2. Otherwise, $K - 2^{\lceil \log_2 K \rceil}$ orthants of the embedding space will not be utilised by the dequantiser. TRUFL has unsupported regions in the decoder as well, but we take the approach of modifying the prior during generation to account for these regions. Furthermore, because the binary vectors are arbitrary, the similarity between words/categories is not reflected in the embedding space as it is in a learned distributed representation.

## C  EXPERIMENTS & IMPLEMENTATION DETAILS

We incorporated the codebases from the following sources:

1. Lippe & Gavves (2020): `https://github.com/phlippe/CategoricalNF`
2. Hoogeboom et al. (2021): `https://github.com/didriknielsen/argmax_flows`
3. Kim et al. (2019): `https://github.com/harvardnlp/compound-pcfg`

### C.1  TOY EXAMPLE: TWO CATEGORY JOINT DISTRIBUTION

In this simple synthetic example, we demonstrate a case in which TRUFL provides an advantage over CatNF and Argmax flow. Our goal is to model a joint probability over 2 categorical random variables, each with 4 categories. This joint distribution is described in Table 5.

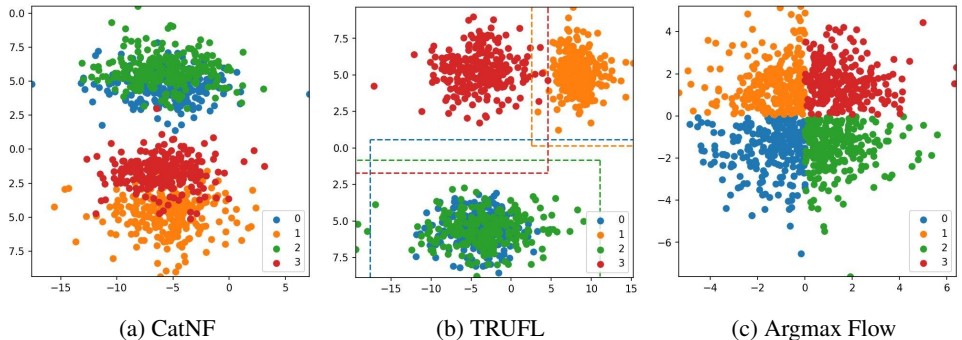

(a) CatNF        (b) TRUFL        (c) Argmax Flow

Figure 5: Embeddings ($q(z_i|x_i)$) of toy example for each method.

Table 5: *Left*: The joint distribution for $x_1$ and $x_2$ for the toy example. *Right*: Log probability over the data of the model, and the breakdown of the ELBO.

| | Cat. | $x_2$ 0 | 1 | 2 | 3 |
|---|---|---|---|---|---|
| | 0 | $\frac{1}{8}$ | 0 | $\frac{1}{8}$ | 0 |
| $x_1$ | 1 | 0 | $\frac{1}{8}$ | 0 | $\frac{1}{8}$ |
| | 2 | $\frac{1}{8}$ | 0 | $\frac{1}{8}$ | 0 |
| | 3 | 0 | 0 | 0 | $\frac{1}{4}$ |

| Method | $\log p(x)$ | $\mathbb{E}\left[\log p(x|z)\right]$ | $\mathbb{E}\left[\log \frac{q(z|x)}{p(z)}\right]$ |
|---|---|---|---|
| CatNF | -2.000 | -1.064 | 0.967 |
| Argmax | -1.915 | -0.687 | 1.263 |
| TRUFL | -1.983 | 0.000 | 2.098 |

Notice that under this distribution, $p(x_1|x_2 = 0) = p(x_1|x_2 = 2)$ and $p(x_2|x_1 = 0) = p(x_2|x_1 = 2)$. For the purposes of this subsection we will say categories 0 and 2 have equal functionality, while categories 1 and 3 are similar in functionality, but not equal.

To model this distribution, we perform variational dequantisation over the discrete variables $x_1, x_2$ using the above methods, and learn a prior over the latent variables. Conditioned on $x_i$, the dequantisation method produces a distribution over a 2 dimensional space — $q(z_i|x_i)$. It is this distribution we plot in Figure 5.

In both CatNF and TRUFL, the distribution $q(z_i|x_i = 0)$ and $q(z_i|x_i = 2)$ are similar. Intuitively, this aligns with the motivation behind the construction of the joint distribution: that the categories are functionally equivalent. However, the dequantisation distribution may not always have this property, as evidenced by the Argmax Flow, due to the way supports are constrained by the orthants. This bimodality in the distributions of these two categories will have to be modelled by the prior, which evidently resulted in a higher log probability in this particular case. In our implementation of Argmax flow for this task, the training eventually resulted in numerical instability. Here we report the best log probability attained before numerical errors occured.

As mentioned in Lippe & Gavves (2020), CatNF can approach deterministic decoding if all other classes are sufficiently far from the target class. However, if several tokens have similar 'function' in the dataset, the corresponding components can be forced to be close together in latent space. When this happens, the closeness in the components again result in decoding becoming non-deterministic. TRUFL allows such components to have minimal and sparse overlapping support but yet have the modes of these components be relatively close. If we consider Figure 5b, we can see that the support and distribution for $x_i = 0$ and $x_i = 2$ overlap the most, while the overlap with $x_i = 1$ and $x_i = 3$ is not as great. Since TRUFL minimises the support for the classes that do not overlap in functionality the TRUFL reconstruction loss is lower in comparison to CatNF.

## C.2   HYPERPARAMETERS FOR LARGE GRAPH COLOURING TASK

## C.3   ARCHITECTURE FOR THE PRIOR IN LANGUAGE MODELLING TASK

We develop our own architecture for this benchmark, replacing only the categorical encoding scheme for each of the benchmarks. The prior is autoregressive both 'in time' and 'in hidden', to use

| Param. | Value |
|---|---|
| batch_size | 128 |
| encoding_dim | 2 |
| coupling_num_flows | 10 |
| optimizer | 4 |
| learning_rate | 3e-4 |

Table 6: New arguments for the LARGE graph colouring task we use in our experiments.

nomenclature from Ziegler & Rush (2019). Autoregression in time is governed by an LSTM, and autoregression in hidden is governed by a MADE model (Germain et al., 2015), conditioned on said LSTM. We use a mixture of 4 logistics per-dimension (Ho et al., 2019) in order to model the multi-modality of the distribution over $z$, which Ziegler & Rush (2019) suggests is prevalent in text data for characters.

