# OpenReview forum: "Learning to Dequantise with Truncated Flows"
_ICLR.cc/2022/Conference — ICLR 2022 Poster_

### Official Review · Reviewer_zxth · 2021-11-01

**Correctness:** 4
**Technical Novelty And Significance:** 3
**Empirical Novelty And Significance:** 2
**Recommendation:** 6
**Confidence:** 4

**Main Review:**

The paper is mostly well written and easy to follow, although I have some reservations about the motivation:

1. I don't think the authors properly motivate the use of the normalizing flow to begin with. Is it really better to have the flow as compared to increasing capacity for m and s? I think this is a very natural question, which is neither posed nor answered in the paper; and that needs answering to better motivate the paper.

2. Part of the motivation is enabling the dequantized variable's dimension to not depend (logarithmically) on the number of categories. It is actually not clear to me that this dependence is particularly harmful. For example, one might expect that categories which are semantically close to each other have corresponding nearby dequantized intervals. However, it might not be possible to achieve this: consider a case with 3 categories, A, B, and C which are though of "semantically equidistant". There is no way to have 3 real intervals satisfying this requirement, and I suspect this might make learning the functions $m$ and $s$ harder. In contrast, if in the same example a 2-dimensional dequantized space was used, one could actually obtain the desired symmetry. I think this comment bears some philosophical resemblance to kernel methods, where the intuition is that simple classifiers might do better on high-dimensional representations. In other words, the fact that one can obtain a one-dimensional dequantization does not automatically imply that one should, and I believe the authors should make a stronger point for this choice.

3. Finally, while dequantization is a sensible approach, I think that it should be at least empirically verified that dequantizing and using a continuous model actually outperforms using a flexible discrete model to begin with. In fairness to the authors though, I think this is more of an issue I have with dequantization itself rather than the paper being reviewed.

As for novelty, the method is novel as far as I am aware.

Finally, I did not find the experiments particularly convincing: while the proposed method does seem to be slightly better than the baselines at the evaluated tasks, the improvements seems small and do not in my view compensate for the issues I have with the motivation. Additionally, only averaged results over runs are reported, and given how close the numbers are along with the lack of error bars, it is hard to assess whether the improvements are actually significant.

Minor remarks:

-In algorithm 1, $\hat{z}_t$ is used in the denominator for the change-of-volume term, but $\hat{z}_t$ is not defined.

-The notation in equation 7 should be changed to either $\tilde{f}(z;m,s)$ or $\tilde{f}(z|m,s)$ to more clearly differentiate between the different types of inputs.

-Table 2 should be referenced in the text in section 5.2.

==========================================================================================================

UPDATE AFTER REBUTTAL

==========================================================================================================

I have increased my score after seeing the authors' updates regarding motivation and the promised ablations regarding point 3 above.

**Summary Of The Paper:**

This paper proposes a dequantization scheme for categorical data, where unlike with ordinal data, element-wise uniform noise cannot be used. The authors propose to encode categorical data into an interval centre and a deviation, and a (pre)dequantized value is obtained by sampling uniformly in the implied real interval. A normalizing flow is then applied to obtain the dequantized variable. The proposed method can thus in principle learn non-overlapping one-dimensional supports for the dequantized variables, enabling lossless dequantization.

**Summary Of The Review:**

While this paper proposes a simple idea for dequantization, I am not convinced the proposed method is needed in the first place, and would need to see much stronger empirical evidence to be convinced otherwise.

---

> ### Author Response · Authors · 2021-11-16
> **Minor remarks addressed, some difficulty understanding the first point.**
>
> Thank you for your comments and detailed feedback. We have taken into account your remarks to improve the clarity of the work and to better motivate TRUFL by highlighting its strengths and flexibility over CatNF and Argmax flow. We hope our response below would sufficiently address your concerns.
> We’ve modified the paper to address your minor remarks. $\hat{z}$ is defined inside the algorithm, in the previous line. It is the inverse of $z$ given a different $g^{-1}$.
> > “I don't think the authors properly motivate the use of the normalizing flow to begin with....”
>
> We have some difficulty understanding this part of the question.
> You are correct in pointing out that there are two additional truncation parameters m and s for each latent dimension. If this is a question about naming our proposed dequantiser a flow, the reason for this is due to the use of an injective reparameterisation trick for gradient estimation.
>
> If this is a question about why not just add more flows to parameterise a more flexible dequantiser, our framework is compatible with that, with the additional flexibility to have a truncated base distribution. This could be helpful especially because it is well known that most commonly used flows do not adapt the tails of the base distribution, and that truncation allows us to learn a lossless dequantiser without pushing the modes away from each other.

---

> ### Author Response · Authors · 2021-11-16
> **TRUFL dimensionality is independent of the number of categories, not strictly less.**
>
> > Part of the motivation is enabling the dequantized variable's dimension to not depend (logarithmically) on the number of categories…
>
> The constraint in the case of Argmax flows is not only that the dequantised variable dimensions depend on the number of categories, but also that it partitions the vectors into “arbitrarily chosen” orthants of the latent space. In your example with 3 categories being semantically equidistant, in a 2 dimensional space (up to 4 categories), the only option is for all categories to be close to the origin. If, however, a 4th category is similar to C, but not to A, for example, this distribution would be forced to be near the origin as well. This highlights the benefit of having a learnable dequantiser such as CatNF and TRUFL, since the latter allows us to learn a more meaningful representation for the latent prior to model.
> TRUFL generalises both CatNF and Argmax flow, which allows us to actually parameterise a dequantiser that has the same dimensionality as the (logarithmic) number of categories. This means TRUFL can approximate Argmax flow as a special case. The benefit of this generalisation is so that for categories that are “semantically close”, TRUFL would be able to reflect that in the latent space, while for categories that do not possess the same functionality it also has the flexibility to draw a sharper decision boundary and admit lossless encoding.
> We have included a toy example in Section 5.1 that hopefully illustrates these two points more clearly. See the common response for more details.
>
> > In other words, the fact that one can obtain a one-dimensional dequantization does not automatically imply that one should, and I believe the authors should make a stronger point for this choice.
>
> We would like to clarify that we do not use single-dimensional dequantisation. Additionally, one benefit of both CatNF and TRUFL is that the latent variable dimensionality is not dependent on the number of categories, and so it can also be _more_ than logarithmic (or linear) in the number of categories. Instead, the dimensionality of the latent variable is a hyperparameter. We have also added a new paragraph (at the end of Section 3.1) explaining that the supports of both CatNF and Argmax Flow can be recovered by specific assignments of $m$ and $s$, as special cases of TRUFL.
>
> > Finally, while dequantization is a sensible approach,...
>
> We understand and appreciate this comment. We are likewise working on methods that might bring the benefits that the continuous model offers while still performing on-par or outperforming a flexible discrete model. We see this as a step in that direction, though not fully arriving at the destination just yet.
> Another motivation for dequantisation + continuous density model over the latents, as we discussed in the introduction, is that while some data might be stored in discrete format, they might have an intrinsic ordinal topology, such as word embeddings and representations of atoms. While modeling discrete data as is using a discrete model is a valid and strong alternative, it restricts the family of models that we can choose from (typically autoregressive models). To argue in favour of dequantisation, it allows us to tap into a larger family of likelihood-based models that can be useful for representation learning of the dequantised embedding which might be beneficial to the overall performance and interpretability. As an example of a performance benefit, WaveGLOW (Prenger et. al. 2018) uses flow-based methods for speech synthesis, allowing for faster sampling (compared to an autoregressive model) by exploiting the parallelism of such methods. The same benefit could be achieved for sampling in categorical data as well.
>
> > the lack of error bars
>
> We have added the error bars to the respective tables. The number of runs for each task are in the captions, and we report the standard deviation following Lippe & Gavves(2020).

---

> > ### Author Response · Authors · 2021-11-18
> > **We hope you had a chance to look at our response.**
> >
> > We hope the updated draft is to your liking.
> > If our response cleared up any prior misconceptions or concerns you had, we hope you will consider increasing your score.

---

> > ### Comment · Reviewer_zxth · 2021-11-22
> > **Discussion**
> >
> > I thanks the authors for their reply to my review.
> >
> > I appreciate the changes in the manuscript, and do agree they better motivate the proposed method. Upon re-reading my review I realize I was unclear with my example: I understand you can recover multi-dimensional dequantizations, I just meant to highlight the point that the dimension required to meaningfully do so might be high enough that the logarithmic requirement from previous work ends up not being an issue.
> >
> > As for the misunderstood question, I was wondering how performance changes as one shifts capacity from the NF and puts that capacity in the $m$ and $s$ networks. Can one get good performance without a flexible flow, or even without a NF altogether, as long as more capacity is added to $m$ and $s$?

---

> > > ### Author Response · Authors · 2021-11-22
> > > **Binary encoding assignment of orthants is another Argmax Flow requirement, TRUFL without additional flow**
> > >
> > > > the dimension required to meaningfully do so might be high enough that the logarithmic requirement from previous work ends up not being an issue.
> > >
> > > Having a dimensionality logarithmic wrt to the number of categories is not the only requirement for argmax flows. The encoding scheme also requires that each category be limited in support to an arbitrarily chosen orthant (this orthant is assigned by binary encoding of the category index, which at times is chosen at random). This creates an additional constraint over the dimensionality of the latent space. In our proposal, we limit support like argmax flow, which allows for deterministic decoding under specific cases, but allow for these supports to be learned.
> > >
> > > >  Can one get good performance without a flexible flow, or even without a NF altogether, as long as more capacity is added to $m$ and $s$ ?
> > >
> > > This is a great question. Without a shift and scale following the truncation, the distributions are all truncated standard logistic distributions.
> > > An alternative formulation would be to fix the truncation but only allow for shift and scale to be learned, since the shift and scales affect the eventual support for the distribution for a given word. We think this will result in a weaker model, but it should be interesting to run an ablation study for the camera ready should this paper be accepted.

---

> > > > ### Comment · Reviewer_zxth · 2021-11-22
> > > > **Discussion**
> > > >
> > > > Thank you for the additional clarification. I have decided to increase my score, given the added motivation and the promise to include ablations against decreasing the capacity or removing the NF while increasing the capacity of $m$ and $s$ upon publication.

---

> > > > > ### Author Response · Authors · 2021-11-22
> > > > > **Thank you!**
> > > > >
> > > > > Thank you for taking the time to understand and clarify with us, and helping us improve our paper.

---

### Official Review · Reviewer_Pdpw · 2021-11-02

**Correctness:** 3
**Technical Novelty And Significance:** 2
**Empirical Novelty And Significance:** 3
**Recommendation:** 6
**Confidence:** 3

**Main Review:**

Learning how to handle discrete data in continuous deep neural networks is an important task, and contributions in this direction are very useful. However, I am a bit concerned with the authors' claim that TRUFL better achieves the two goals mentioned in their introduction: (i) easily learnable, and (ii) lossless. First, it is unclear whether TRUFL is more "lossless" than CatNF in practice. While TRUFL uses truncation to zero out the probability mass beyond a bounded range, there could still be major overlap between the latent space of different categories. And compared to that, truncating the low-probability regions might be less effective in making the dequantization lossless. To make the dequantizations more lossless, another simple choice could be to make distributions on z farther away from each other (Figure 1 CatNF, making the three gaussian farther away from each other).

In fact, I can imagine in certain cases we don't need lossless dequantization, as some variation of the data might be irrelevant to a task.

While the proposed TRUFL model may still perform lossy dequantization, according to the claim in the introduction, the benefit of TRUFL could be it's easily learnable. However, due to the similarity with CatNF and the additional need for rejection sampling, it is not very clear to me why TRUFL can be learned easily. To summarize, it is unclear to me why truncating the latent space could simplify the learning problem itself.

Since the proposed approach uses rejection sampling, it would be nice to also compare the inference time of different models, just to see how much the sampling procedure influences the model's overall efficiency.

**Summary Of The Paper:**

This paper proposes a Flow model called TRUFL designed for discrete data. The main selling point of the proposed model is that it can handle discrete data better than other dequantization schemes. The authors target two key difficulties of this dequantization problem, which are (i) making the dequantization lossless, and (ii) allowing the dequantizer to be learned easily. Building on the Categorical Normalizing Flows, the authors propose to truncate the dequantized latent space to make the latent space of different input categories less correlated. To compute the probabilities w.r.t. the truncated latent space, the authors use rejection sampling to approximate these probabilities.

**Summary Of The Review:**

I tend to vote for rejection because it is unclear to me why truncating the latent distribution could necessarily improve the dequantization quality. Although the authors provide two explanations in the introduction, they are not very well justified. Apart from the truncation technique, the proposed model doesn't seem to differ too much from existing approaches.

---

> ### Author Response · Authors · 2021-11-16
> **TRUFL does not need rejection sampling during training, and can learn has the flexibility to find a happy medium between CatNF and Argmax Flow**
>
> We thank you for your feedback and questions. Overall the reviewer believes our work is both technically and empirically novel and significant, and correct overall. We will try to address the few issues that require further clarification. These include the use of rejection sampling and the losslessness (or lossiness) of TRUFL.
>
> Rejection Sampling:
> * During training, rejection sampling does not come into play. Instead, we use the reparametrisation trick to estimate the gradient of the dequantiser, as described in Section 3.1, equations 4-7. This is unlike the prior work of Polykovskiy & Vetrov, 2020, which requires rejection sampling during training.
> * We use rejection sampling at test time, if we want to sample from the model (by restricting the support of the prior, as described in Section 3.2).This is done for the sampling tasks in the experiments (graph colouring and molecule generation), which appears to increase the proportion of valid samples we get from the model.
>
> Lossless vs Lossy dequantisation:
> * As our method generalises both CatNF and Argmax Flows, it has the flexibility to find a happy medium between the two in the degree of lossiness it can achieve.
>
> To clarify, our method _can_ learn the truncation boundaries. If these boundaries result in no overlapping supports, then the quantisation is lossless. In the cases where we do _not_ need losslessness, the model may have learned overlapping truncation boundaries, and so we can also account for those situations.
>
> > “it is unclear whether TRUFL is more "lossless" than CatNF in practice” and “another simple choice could be to make distributions on z farther away from each other”
>
> We have included a toy example (Section 5.1) hopefully demonstrating certain situations (particularly when categories have overlapping semantics, as you have pointed out) where the distributions may be close. In these cases, moving the distributions on z further apart may not be an option. The truncated distributions allowing for limited support reduce the reconstruction loss, and increase the overall log-likelihood. In this particular toy setup, TRUFL does have a low reconstruction loss, i.e. more lossless. However, you are right that even in these cases, you can still move the distributions on z farther away, but the prior p(z) will have to be sufficiently powerful to model these multiple modes.
>
> To summarise, TRUFL can be trained via the reparameterisation trick, which does not need rejection sampling. Rejection sampling is only used at test time to restrict the support of the prior, which increases validity of samples in some experiments. Further, as per the last paragraph of Section 3.1, by setting m and s to appropriate values, we can recover each of these variational dequantisation schemes. This allows TRUFL to approximate Argmax flow (to be more lossless) or CatNF (to have shared support). We hope this clarifies any doubts you may have had.

---

> > ### Author Response · Authors · 2021-11-18
> > **We hope you had a chance to look at our response.**
> >
> > We would appreciate if you would consider increasing the score if our response cleared up any prior misconceptions or concerns you had.
> > Please let us know if we addressed your concerns and questions.

---

> > ### Comment · Reviewer_Pdpw · 2021-11-27
> > **Thanks for the responses**
> >
> > The authors' responses have clarified my concern regarding the training efficiency (since rejection sampling is not used in the training time).
> >
> > > even in these cases, you can still move the distributions on z farther away, but the prior p(z) will have to be sufficiently powerful to model these multiple modes
> >
> > This seems to suggest that the truncation is making the prior more powerful, and resembles using more expressive priors. However, since the truncation is only used at test time, it seems to me that truncating z cannot have an equivalent effect with using more expressive priors. Also in the original review, "make distributions on z farther away from each other" does not necessarily make the prior more expressive, since they can still be e.g. Gaussians. Please correct me if I am wrong, but I am still not sure about the difference between truncating z and making distributions on z farther away from each other.

---

> > > ### Author Response · Authors · 2021-11-27
> > > **May have given wrong impression, not suggesting truncation is making the prior more powerful.**
> > >
> > > We apologise if we have given that impression with our response, but this is not what we meant by “the prior p(z) will have to be sufficiently powerful”.
> > >
> > > In order for CatNF to approach deterministic decoding, the distributions for each category will have to be moved sufficiently far apart. This will result in a highly multi-modal aggregate posterior, which the prior will have to model. This will require a prior that is  ‘sufficiently powerful’, and such a prior may be difficult to learn or parameterise.
> > >
> > > In contrast, truncated distributions can be _less_ multi-modal. Consider 2 categories that map to 2 truncated logistic distributions along the real line: one with support from $(-\infty, 0)$, and one from $(0, \infty)$. Both categories will still have deterministic decoding, but in this case, the corresponding aggregate posterior can be modelled by a single logistic distribution.
> > >
> > > In other words, a simpler prior may work just as well in cases of truncated posteriors, while still offering deterministic decoding.

---

> > > > ### Author Response · Authors · 2021-11-28
> > > > **Discussion period coming to an end**
> > > >
> > > > As the discussion period is coming to an end, do let us know if we have cleared up all of your previous doubts.
> > > > Do let us know if you have any remaining concerns. We would appreciate if you would consider increasing the score if you found the changes and our response useful.

---

> > > > ### Comment · Reviewer_Pdpw · 2021-11-29
> > > > **Thank you for the detailed response**
> > > >
> > > > The authors clarified some of my main concerns. Still, there are some aspects of the method that I hope could be clarified/addressed:
> > > >
> > > > - If the prior for different categories are too "close" to each other, truncation will be hard to perform and could potentially "cut off" part of the high-probability region in the latent space. Will this affect the performance? If yes, by how much.
> > > >
> > > > - Making the priors for different categories far apart from each other does not seem to be very hard. So the reason truncation works might be that it is more strict than what is done by CatNF. It would be helpful to justify this explicitly with some experiments.

---

> > > > > ### Author Response · Authors · 2021-11-29
> > > > > **Unsupported prior density and Multi-modality in the latent space**
> > > > >
> > > > > > If the prior for different categories are too "close" to each other, truncation will be hard to perform and could potentially "cut off" part of the high-probability region in the latent space. Will this affect the performance? If yes, by how much.
> > > > >
> > > > > We have the following interpretations of your question. Please let us know if we sufficiently address your concerns.
> > > > >
> > > > > 1. If you mean that the mode of this conditional distribution may be ‘outside’ of the truncated region of the dequantiser $q(z|x)$, this is renormalised by $s$ in Equation (7). So in *training* we do not cut off any mass from $q$, meaning there is no likelihood “leaking” out of the support. Sampling from $q$ during training is performed by inverse CDF transform, so there is no *computational efficiency* issue either.
> > > > >
> > > > > 2. If you mean that the unioned support of the posterior may not cover high-probability regions of the prior, then: yes, this is possible. In practice, there is usually an overlap between the different categories, which can be seen in our newly added toy example in Section 5.1. The KL term indirectly ensures that supports are closer to the modes of the prior, though this constraint is not a hard one.
> > > > >
> > > > > Theoretically though, if we renormalise the support with respect to the unioned support of the posterior, this problem can be alleviated. We do have an idea of what this gap is, by way of the bound seen in Equation (8). In the graph colouring experiment, we can estimate how much of a loss in terms of nats we incur: with an acceptance rate of 86%, this corresponds to a gap of ~0.15 nats (see: Section 5.2, Rejection rate of TRUFL). This means that sampling at *test* time is *computationally efficient* and that the gap of likelihood evaluation is small. This gap may differ from task to task, and in other tasks this estimation can be harder to compute. But we stress that the values we reported are all lower bounds on the true likelihood. If said renormalisation is performed, our numbers would always be higher in expectation. This is in addition to the variational gap caused by dequantisation.
> > > > >
> > > > >
> > > > > > Making the priors for different categories far apart from each other does not seem to be very hard. So the reason truncation works might be that it is more strict than what is done by CatNF. It would be helpful to justify this explicitly with some experiments.
> > > > >
> > > > > Regarding the difficulties in learning well separated modes using a flow-based prior, there has been some evidence that suggests it is hard both in theory and in practice, as the prior might have very different topological properties (such as connectedness) than the target (data) distribution. See references [1-5] below.
> > > > >
> > > > > Our toy example in Section 5.1 also seeks to demonstrate that the general behaviour of both encoding schemes is to push distributions of categories with similar *functions* close together. Indeed, the truncation helps in situations like this by truncating support where the functionality of the categories do not overlap, allowing for a better reconstruction loss (likelihood term), so in that sense, it is more ‘strict’. This is also evident in that toy example.
> > > > >
> > > > > 1. Relaxing Bijectivity Constraints with Continuously Indexed Normalising Flows, 2019
> > > > > 2. Augmented Normalizing Flows: Bridging the Gap Between Generative Flows and Latent Variable Models, 2020
> > > > > 3. Vflow: More expressive generative flows with variational data augmentation, 2020
> > > > > 4. SurVAE Flows: Surjections to Bridge the Gap between VAEs and Flows, 2020
> > > > > 5. Variational Inference with Continuously-Indexed Normalizing Flows, 2021

---

### Official Review · Reviewer_QH7t · 2021-11-02

**Correctness:** 4
**Technical Novelty And Significance:** 3
**Empirical Novelty And Significance:** 3
**Recommendation:** 6
**Confidence:** 4

**Main Review:**

**Pros:** The paper is very well written and the method is explained clearly, concisely and in appropriate detail. The use of examples, figures and algorithm boxes at appropriate places makes the paper easy to follow and understand. Furthermore, the problem considered in the paper is a valid and interesting problem for modelling discrete data using NFs. Multiple papers have come out in the past year or so addressing this problem, each with their own sets of solutions (and restrictions) and the present paper proposes a new method that alleviates some of the drawbacks of CatNFs and Argmax flows. The experimental analysis by the authors consider a diverse set of problems from graph coloring, molecular generation, and language modelling with reasonable performances.

**Cons:** I found the empirical results to be mixed for TRUFL as compared to Argmax flows and CatNF. While, this in itself not a huge issue, I am interested to understand what might explain this considering that the method basically tries to address the drawbacks of both these approaches. I also wonder if it might be possible to use synthetic examples to bring out these advantages for TRUFL.

**Summary Of The Paper:**

The paper presents a new approach for dequantization ie embedding discrete data in a continuous space using variational inference and truncated flows called TRUFL. Unlike previous approaches, TRUFL aloows the dequantization layer to have a learnable truncated support. The authors perform several experiments to demonstrate the advantages of the proposed method to Categorical Normalizing Flows (CatNF) and Argmax Flows.

**Summary Of The Review:**

The paper is well written and explored. The empirical analysis is reasonable. Overall, I found the paper quite interesting.

---

> ### Author Response · Authors · 2021-11-16
> **Thank you. We added a 2 subsections to the paper to address your comments.**
>
> Thank you for your comments. We are glad that you found the paper well written. We have tried to improve the narrative of the paper further to contrast our method against Argmax flow and CatNF. Specifically, TRUFL generalises the two as it has the flexibility to approximate either one as a special case, which allows it to learn a dequantisation scheme closer to the preferable one depending on the problem. See end of Section 3.1 for more details.
>
> Additionally, having looked at the comments from some of the other reviewers, we have tried to design a toy example where the TRUFL has advantages over both the Argmax Flows and CatNF. We hope you will find the addition useful in understanding the pros and cons of not just TRUFL, but also the CatNF and Argmax Flows. Please see the common response for a more detailed description of the new synthetic experiment.

---

> > ### Author Response · Authors · 2021-11-18
> > **We hope you had a chance to look at our response.**
> >
> > We would appreciate if you would consider increasing the score if you found the changes and our response useful.
> > Please let us know if we addressed your concerns.

---

### Author Response · Authors · 2021-11-16
**Thank you for your comments and feedback. We've updated the paper.**

After reading all of the comments and feedback from the reviewers, we added a new synthetic experiment in Section 5.1, hoping to address some of the common questions and make clear the motivations of TRUFL. Also, we’d like to thank the reviewers for their questions that allow us to craft a toy example that, hopefully, builds intuition. We summarise the details of the experiment setup and  the interpretation here below. For visualisation, please see Figure 3 of the updated manuscript.

This toy example demonstrates a scenario where categories have equal or similar functionality in a joint distribution. Distributions of these categories tend to be close to each other, in cases of CatNF and TRUFL, but due to the construction of Argmax Flow, this does not happen in that case, leaving the prior to model the bimodality, and log probability is affected. Additionally, when the categories are close to each other, CatNF’s ability to approximate lossless decoding is diminished, while TRUFL can allow the local support to overlap, while having flexibility to learn supports for unrelated categories to be non-overlapping. This results in a better reconstruction loss for TRUFL.

Additionally, we have added a paragraph at the end of Section 3.1, to clarify that our specific instantiation of TRUFL (with the simple inverse CDF reparameterisation) does generalise both Argmax flow and CatNF. This extra flexibility allows TRUFL to approximate either one of the two depending on which one is preferable for the task.

---

### Decision · Program_Chairs · 2022-01-20

**Decision:**

Accept (Poster)

**Comment:**

The paper proposes a variational dequantization method for categorical data, based on flows with learned truncated support. The problem has been studied before, but the paper makes it clear how the proposed method differs from existing ones. The method is empirically evaluated on a large variety of diverse tasks.

The reviews were initially borderline. In general, the reviewers did not identify major quality of technical issues with the paper, and appreciated the clarity of writing. On the other hand, the reviewers were not fully convinced by the motivation or the empirical performance of the proposed method. After discussion with the authors, some concerns were allayed (especially regarding motivation) and all three reviewers decided to recommend weak acceptance.

Seeing as there are no major technical or quality issues with the paper, and the paper is clearly written and well executed, I'm leaning towards recommending acceptance, although some doubts remain about the significance of the contribution.